# Major chemical carcinogens and health exposure risks in some therapeutic herbal plants in Nigeria

**Raymond Limen Njinga** \*, **Ayodele Philip Olufemi, Adebiyi Samuel Adebayo**

University of Medical Sciences, Ondo City, Ondo State, Nigeria

\* rnjinga@unimed.edu.ng

## Abstract

People of all ages and genders utilize herbal medicine to treat varieties of problems all around the world. The accumulation of Cd and Cr in therapeutic herbs (*Adansonia digitata*, *Psidium guajava*, *and Carica papaya*) can lead to a variety of health complications. These leaf extracts are used to treat varieties of ailments, including cancer, in the northern Nigerian states of Borno, Jigawa, and Kano. The researchers employed high-resolution continuous source atomic absorption spectrometry. The statistical parameters such as mean, range, minimum and maximum were computed along with one-way analysis of variance (ANOVA) to assess activity concentrations of Major Chemical Carcinogens (MCCs) in the herb extracts from the three states. The result demonstrated substantial statistical variation in the concentration of Chromium between groups with *C. papaya* (F = 190.683, p = 0.000), *P. guajava* (F = 5.698, p = 0.006), *A. digitata* (F = 243.154, p = 0.000). The post hoc test revealed that the C. papaya and A. digitata observed concentrations were statistically significant across the three states (p = 0.000). It was observed that there is no statistically significant difference between concentrations of the extracts between Kano and Borno states for *P. guajava* (p = 0.686). For Cd, the one-way ANOVA showed significant statistically variation in the concentration between groups with *C. papaya* (F = 77.393, p = 0.000), *P. guajava* (F = 4.496, p = 0.017), *A. digitata* (F = 69.042, p = 0.000). The post hoc test with multiple comparisons revealed that the activity concentration of all extracts was statistically significant across the three states (p<0.05). The target risk quotient (THQ) for Cd was more than unity in *A. digitata* and *C. papaya*, except for *P. guajava* from Borno State. The probable cancer risk was observed for consumption of plant extracts as a result of Cr and Cd.

## Introduction

In most parts of the world today medicinal plants have been recognized to offer three primary types of advantages: (i) Health benefits to people who consumed them, (ii) financial advantages to those who cultivate, prepare, and sell them; and (iii) job opportunities to people who administered them to the populace [1]. Medicinal plant treatment is independent of any age group and sex. The emphasis on research in medicinal plants treatment is in increasing in recent

**Data Availability Statement:** All relevant data are within the paper.

**Funding:** No support from anywhere.

**Competing interests:** The authors have declared that no competing interests exist.

times and a large number of scientific outputs to substantiate the importance of these plants in different traditional systems all over the world are on the rise [2, 3]. The ecosystems of northern Nigeria are rich in a diverse range of valuable tree species that produce non-wood forest products. Fruit trees that are native to the area are particularly significant for the nutritional quality of the rural inhabitants as well as their earnings [4, 5]. A deeper understanding of the possible applications of these species, as well as the limits that come with the transformation and commercialization of their products would be beneficial to their promotion.

Food insecurity affects rural households in Sub-Saharan Africa. The global economic crisis and rising orthodox pharmaceutical prices are wreaking havoc on practically every livelihood, particularly in rural areas. With an increasing rural population and a drop in the number of hospitals, health centers, and doctors/health workers, it is necessary to discover alternative or complementary sources of herbal traditional therapy to meet the expanding demands in rural areas. Natural resources abound in Africa. A great diversity of medicinal herbs can be found in the northern Nigerian ecosystems. Unfortunately, extension agencies have disregarded the utility and commercial worth of these trees in favor of over-promoted exotic fruits. In recent decades, there has been an increasing interest in herbs as a source of revenue and poverty reduction, and as a result, non-qualified people can be found giving medical plants carelessly all over the world [6]. Developing measures for a sustainable harvest through joint/or participatory registered traditional herbal management is one strategy to improve livelihoods while minimizing biodiversity loss and over-exploitation of these herbs. Indigenous fruit trees like *Psidium guajava* and *Carica papaya* play a vital role in rural households' diets, and as a result, there has been greater recognition of their importance [7]. These fruits are eaten raw and can be used in a variety of dishes as veggies or spices. These under-evaluated products contribute to improved rural nutrition, family income, and the country's economy. To become commercially important, these herbal plants must be used outside national borders, which necessitates improved government regulations as this work aims to achieve and the first step is to build a deeper understanding of the possible utilization of diverse species and the quality.

In Northern Nigeria (Kano, Jigawa, and Borno States) the consumption of herbal remedies for therapeutic reasons is high and emphasis is placed on the leaves of *Adansonia digitata*, *Psidium guajava* and *Carica papaya* [8]. A lot of studies have shown that papaya leaves contain antioxidants, digestive enzymes and other nutritional factors which protect against the development of cancer [9–11]. According to [12], the leaf extracts of A. digitata, P. guajava and C. papaya can provide potent cures for various types of ailments in humans. The study of [13] shows that the leaf extracts of P. guajava contain antioxidants, anti-inflammatory agents, antibacterial and tannins that have health benefits. They demonstrated that the leaves, when brewed to make tea, contain vitamin C and flavonoids like quercetin. Research findings by [14] also revealed that the leaf extracts may serve as an alternative cure for diarrhea as shown in Table 1, and may be used to lose weight. Further study [15] show that *P. guajava* contains high fibre content that is used to manage diabetes. Other studies [16–18] indicate that regular drinking of leaf juice may reduce low-density lipoprotein (LDL) cholesterol and triglycerides without any side effects and that they may be used to treat patients with gastric (stomach) cancer as shown in Table 1.

*A. digitata* is a plant that is found almost in every country in Africa and some of the species were exported and cultivated in India [14, 19]. A study shows that the fruits of this tree are exotic and are rich in nutrient profile. Generally, in Nigeria, the leaves, bark, and seeds of this plant (*A. digitata*) are used to treat numerous types of sicknesses such as malaria, tuberculosis, fever, microbial infections, diarrhea, anemia, dysentery, and toothache as indicated in Table 1.

Many research studies have established that Cd and Cr have adverse effects on humans and are highly carcinogenic at low levels. These metals may enter the environment and food chain

**Table 1. Therapeutic application of the three herb extracts in Nigeria [8].**

| Name | Family | Part used | Method of consumption | Therapeutic application |
|------|--------|-----------|----------------------|------------------------|
| *C. papaya* | Caricaceae | Leaves | Tea, leaf juice extract, food | Cancer, diabetes, skin ailments, malaria, ulcer |
| *P. guajava* | Myrtaceae | Leaves | Tea, leaf juice extract, food | Diarrhea, obesity, diabetes, gastric and prostate cancer |
| *A. digitata* | Malvaceae | Leaves | Tea, leaf juice extract, food | Malaria, tuberculosis, fever, microbial infections, diarrhea, anemia, dysentery, toothache |

via direct or indirect ingestion [20, 21]. However, hazardous heavy metal poisoning has been shown to have a greater impact on public health in underdeveloped nations than in developed countries [22]. Nigeria's fast industrialization and urbanization have contributed to, and continue to contribute to, serious environmental concerns [23–25]. This demonstrates that, as a result of growth, the ecosystem and food safety are in peril, as companies release their wastes into the environment indiscriminately and without or with little treatment [20]. In this sense, environmental problems and associated health hazards to the people are more pronounced in developing nations like Nigeria, where environmental safety laws have not been adhered to. Therefore, this work aims to evaluate the concentration level of major chemical carcinogen in some therapeutic herbal plants in northern Nigeria (Kano, Jigawa, and Borno states) and assessed its probable heath risk impact. The aim were achieved by evaluating the statistical parameters such as mean, range, minimum and maximum along with one-way analysis of variance (ANOVA) to assess activity concentrations of MCCs in the herb extracts from the three states. The estimated daily intake (EDI), target hazard quotient (THQ), hazard index (HI), and target cancer risk (TCR) were used to evaluate the possible health concerns related to the usage of these plant extracts.

## Materials and methods

### Identification of herbs

The plants were identified by the University of Medical Sciences, Department of Biological Sciences, Plant Biology and Biotechnology unit with the following herbarium numbers;

1. *UNIMED P.B.T.H No 0009 (A. digitata)*

2. *UNIMED P.B.T.H No 0010 (C. Papaya)*

3. *UNIMED P.B.T.H No 0011 (P. guajava)*

**C. Papaya** leaves were identified as being relatively large, with an average diameter of 5070 centimeters and length of 18–90 cm, and are broad, flat, and deeply palmate [26]. The rough green leaves are thin, flexible, and have prominent yellow veins that span all 5–9 lobes.

**P. guajava** was identified as an evergreen shrub (small tree) that can reach a height of 3–10 m [27]. It is observed to have a shallow root system and produces low branches at the base and suckers from the roots. The trunk is thin, 20 cm in diameter, and covered with a smooth green to reddish-brown bark that comes off in thin strips [28]. The leaves are oblong to oval in shape and average 7–15 centimeters long and 3–5 centimeters wide [29]. The leaves grow in an opposite arrangement, which means that two leaves grow in the same location on either side of the stem and have short petioles, or stems, that connect the leaf to the stem.

**A. digitata** was identified as a massive deciduous tree, growing up to 20–30 m tall and reaching up to 2–10 m in diameter in the matured stage. The trunk is often of great circumference. The bark is smooth, reddish-brown to gray in color, soft, and with longitudinal fibers [30]. The tree is highly branched and produces a large lateral root system up to 50 m from the trunk. The root tips are often in the form of tubers. But the main roots of older trees are

relatively shallow and rarely extend beyond 2 m. Young leaves on trees are usually simple. Adult trees begin each season to produce single leaves followed by 2–3 leafy leaves; mature leaves (20 cm in diameter with about 5–9 leaflets) appear later. The inflorescence of baobab trees is a single flower, located in the leaf axils near the tips of breeding branches. The flowers are white, large, pendulous, and solitary or paired in the leaf axils, showy.

## Samples collection

The plants were collected haphazardly from Kano, Jigawa and Borno with no special permit required since they are readily available almost everywhere in the regions. A total of one hundred and forty-seven (147) leaf samples were collected with the permission of the farmers as follows:

- Kano, 18 samples (2 kg each) of *C. papaya* (n = 18) were collected at random from 18 different locations while 15 samples (2 kg each) of *P. guajava* (n = 15) were taken at random from 15 different locations and 16 samples of 2 kg each of A. digitata (n = 16) were randomly sampled in 16 different locations.

- Jigawa, 18 samples of 2 kg each of *C. papaya* (n = 18) were arbitrarily sampled from 18 different locations while 15 samples (2 kg each) of *P. guajava* (n = 15) were randomly obtained from 15 different locations and 16 samples of 2 kg each of *A. digitata* (n = 16) were randomly sampled from 16 different locations.

- Borno, 18 samples (2kg each) of *C. papaya* (n = 18) were willy-nilly sampled from 18 different locations while 15 samples (2kg each) of *P. guajava* (n = 15) were obtained arbitrarily from 15 different locations, and 16 samples of 2kg each of A. digitata were (n = 16) sampled randomly from 16 different locations.

Distilled water was used to wash the samples thoroughly to eliminate dirt, dust, and other possible parasites, and then they were properly rinsed with de-ionized water. The rinsed samples were dried under shade and stored in clean and dried perforated leather bags. The prepared samples were transported to the laboratory for further processing.

While in the laboratory, the prepared samples were heated in a multivalve 3000 microwave system for further drying at a temperature of 60 to 65 ˚C for 45 min. The dried samples were grinded using a mortar and pestle to powder form. One gram of dry weight of each of the powdered samples was taken and digested using Aqua Regia; which is a mixture of concentrated nitric acid ($HNO_3$) and concentrated hydrochloric acid (HCl). The digestion was done to analyse the Cd and Cr using Atomic Absorption Spectroscopy. Descriptive data analysis was performed with SPSS version 16.0 statistical software. The statistical parameters such as mean, range, minimum and maximum were computed along with one-way analysis of variance (ANOVA) to assess activity concentrations of MCCs in the herb extracts from the three states.

## Health Hazard Assessments (HHA)

**Estimated Daily Intake (EDI).** The equations and exposure properties developed by the United States Environmental Protection Agency (USEPA) for health hazard assessment were adopted in the assessment of probable health conditions in the consumption of herbal plants. HHA establishes a relationship between pollutant loads and human health. The estimated daily intake (EDI) of individual metal was computed by substituting the respective exposure parameters (Table 2) and the average concentrations of the measured heavy metal into Eq (1)

**Table 2. Exposure parameters for health risk assessment through ingestion pathway [32–35].**

| Parameters | Unit | Value |
|---|---|---|
| Concentration of metal in herbal plant ($C_M$) | mg/kg dry weight | Study data |
| Exposure rate ($E_f$) | day/year | 365 |
| Exposure period ($E_D$) | years | 65 |
| Average vegetable consumption ($F_{IR}$) | g/person/day | 240 (for low fruit and vegetable intake) |
| Concentration conversion factor ($C_f$) | | 0.085 (Fresh to dry vegetable weight) |
| Body Weight ($B_W$) | kg | 70 (Adults) |
| Average exposure time ($T_A$) | days | 23725 |
| Oral reference dose ($R_fD$) | (mg/kg/day) | Cd:0.001; Cr:0.003 |
| Oral cancer slope factor in (OCSF) | (mg/kg/day)$^{-1}$ | Cd:0.001 |
| | | Cr:0.003 |

as expressed by [31–33].

$$EDI = \frac{E_f \times E_D \times F_{IR} \times C_M \times C_f}{B_W \times T_A} \times 0.001 \qquad (1)$$

The units, values, and interpretation of each parameter contained in Eq 1 are presented in Table 2.

**Target Hazard Quotient (THQ).** The USEPA has established the THQ to assess the potential risk of consuming toxins, including several heavy metals. The lifetime risk is assessed using the oral reference dose as the upper safe limit for the hazardous substance and regular intake over a lifetime in this estimation of concern. The THQ is employed to evaluate the health risk due to the consumption of the herbal plants possibly polluted by toxic metals (Cd, Cr) around Kano, Jigawa, and Borno States of Nigeria, THQ was estimated using Eq 2:

$$THQ = \frac{EDI}{RfD} \qquad (2)$$

where RfD is the oral reference dose (Table 2). Long-term exposure to the MCCs can have a carcinogenic, central and peripheral nervous system, and circulatory effects. For humans, typical risks associated with excess exposure to the MCCs are as follows [36];

i. Acute exposure to Cd ≤ day may result in pneumonitis (lung inflammation) and chronic exposure ≤ months may result in lungs cancer, osteomalacia (softening of bones),

ii. Acute exposure to Cr ≤ day may result in gastrointestinal, hemorrhage (bleeding), hemolysis (red blood cell distribution) acute renal failure and chronic exposure ≤ months may result in proteinuria (excess protein in urine, kidney damage) pulmonary fibrosis (lung scarring) and lung cancer.

**Hazard Index (HI).** The specific health risk of heavy metals from consuming contaminated medicinal plants kinds is cumulative and is expressed by the hazard index (HI). As a result, the HI of the different metals studied was determined using Eq (3).

$$HI = \sum_{n=i}^{i} THQ_n; \quad i = 1, \ 2, \ 3, \ \ldots, \ n \qquad (3)$$

HI values below unity (HI < 1) signifies no significant risk of non-carcinogenic effects while HI values above unity (HI > 1) imply the probability of non-carcinogenic health risk occurring

in the exposed population [35, 37]. whereas an HI greater than 10.0 indicates a substantial chronic health impact [35, 38].

**Target Cancer Risk (TCR).** The cancer risk associated with heavy metal ingestion in the medicinal plant, such as Cd and Cr, was calculated using Eq (4) as follows:

$$TCR = \sum_{n=i}^{i} CR_i \; ; \; i = 1, \; 2, \; 3, \; ..., \; n \tag{4}$$

where

$$CR = EDI \times OCS_F \tag{5}$$

where CR stands for lifetime cancer risk due to specific heavy metal intake, EDI stands for estimated daily metal ingestion of the population in mg/day/kg body weight, and OCSF stands for oral cancer slope factor in (mg/kg/day)$^{-1}$, and n stands for the number of heavy metals considered for cancer risk calculation. For the CR calculation, the $OCS_F$ values were 0.5 for Cr and 0.38 for Cd in (mg/kg/day)$^{-1}$.

## Results and discussion

### The metal concentration in Cd and Cr

The concentration of Cd and Cr in the herb extracts from the three states is displayed in Tables 3 and 4. The mean value (±SD) of Cr across the three states (Table 3) ranged from 0.0063 ±0.0004 mg/kg to 0.011±0.0012 mg/kg for *C. papaya*, 0.0225±0.0034 mg/kg to 0.0282±0.0017 mg/kg for *P. guajava* while for *A. digitata* it ranged from 0.009±0.0005 mg/kg to 0.0081 ±0.0003 mg/kg. These results show that the mean concentration of Cr in the herbal extracts was below the permissible limit of 1.30 mg/kg given by WHO [39] at all study sites during the entire study period. As observed, the maximum value of 0.0357 mg/kg of Cr across the three states (Table 3) was observed in *P. guajava* in Kano state while the minimum value of 0.0058 mg/kg was observed in *C. papaya* in Borno state. Most of the smaller industries in the northern part of the country situated in Kano state could be the reason for the observed high concentration value of Cr in Kano. Furthermore, the discharge of wastes from these industries which in turn contaminate the groundwater, which plants used for transpiration of nutrients could also contribute to the high value of Cr in Kano [40, 41]. High levels of pollution with Cr could result in adverse biological effects on humans and some ecological species [22].

The mean value (±SD) of Cd across the three states (Table 4) ranged from ranged from 0.0051±0.00122 mg/kg to 0.0087±0.00008 mg/kg for C. papaya, 0.0082±0.01326 mg/kg to 0.0176±0.0044 mg/kg for *P. guajava* while for *A. digitata* it ranged from 0.0077±0.00011 mg/kg to 0.0092±0.00032 mg/kg. These results show that the mean concentration of Cd in the herbal extracts was below the permissible limit of 0.02 mg/kg given by WHO at all study sites during the entire study period [42]. Jigawa state had the lowest and highest concentration of Cd with the value of 0.0043 and 0.056 mg/kg respectively obtained in *P. guajava*. Studies have shown that Cd found in plant leaves may be a result of soil-to-plant high rate transfer factor [43]. Cd accumulates in many organs throughout one's lifetime as it is ingested. This research revealed that humans are mostly exposed to ingestion of the medicinal plant's leaf extracts.

The difference in the concentration of the analyzed metals in the medicinal plants is displayed in Tables 5–8. Table 5 showed there was a statistically significant difference in Cr concentration between groups as shown by the one-way ANOVA with *C. papaya* (F = 190.683, p = 0.000), *P. guajava* (F = 5.698, p = 0.006), *A. digitata* (F = 243.154, p = 0.000). Post hoc test (Table 6) revealed that the *C. papaya* and *A. digitata* observed concentrations were statistically significant across the three states (p = 0.000). However, no statistically significant difference

**Table 3. Descriptive statistics of the activity concentration of Cr in the medicinal plants from the three states.**

| | | Mean ± SD (mg/kg) | 95% Confidence Interval for Mean | | Minimum (mg/kg) | Maximum (mg/kg) |
|---|---|---|---|---|---|---|
| | | | Lower Bound (mg/kg) | Upper Bound (mg/kg) | | |
| C. papaya | Kano | 0.0082±0.0003 | 0.0081 | 0.0084 | 0.0078 | 0.0086 |
| | Jigawa | 0.011±0.0012 | 0.0104 | 0.0116 | 0.0099 | 0.0125 |
| | Borno | 0.0063±0.0004 | 0.0061 | 0.0064 | 0.0058 | 0.0069 |
| | Total | 0.0085±0.0021 | 0.0079 | 0.0091 | 0.0058 | 0.0125 |
| P. guajava | Kano | 0.0232±0.0079 | 0.0189 | 0.0276 | 0.0148 | 0.0357 |
| | Jigawa | 0.0282±0.0017 | 0.0272 | 0.0292 | 0.0256 | 0.0312 |
| | Borno | 0.0225±0.0034 | 0.0206 | 0.0243 | 0.0178 | 0.0278 |
| | Total | 0.0246±0.0056 | 0.0230 | 0.0263 | 0.0148 | 0.0357 |
| A. digitata | Kano | 0.0081±0.0003 | 0.0079 | 0.0083 | 0.0074 | 0.0086 |
| | Jigawa | 0.0062±0.0002 | 0.0061 | 0.0063 | 0.0059 | 0.0066 |
| | Borno | 0.009±0.0005 | 0.0088 | 0.0093 | 0.0083 | 0.0098 |
| | Total | 0.0078±0.0012 | 0.0074 | 0.0081 | 0.0059 | 0.0098 |

were observed between concentrations of the extracts between Kano and Borno states for P. guajava ($p = 0.685$).

Table 7, shows the one-way ANOVA result for Cd in the plants extract. The result revealed statistically significant difference in Cd concentration between groups with *C. papaya* ($F = 77.393$, $p = 0.000$), *P. guajava* ($F = 4.496$, $p = 0.017$), A. digitata ($F = 69.042$, $p = 0.000$). Post hoc test with multiple comparison (Table 8) revealed that the activity concentration of all extracts was statistically significant across the three states with ($p < 0.05$).

## Health risk assessment of MCCS

The estimated daily intake (EDI) and maximum tolerable daily ingestion (MTDI) of heavy metals in the extracts of the study area are presented in Table 9. The EDI of heavy metals for an adult as a result of consumption of 240 g/day of the herbs were observed for Cr in C. papaya to be 2.40E-03, 3.20E-03, & 1.82E-03 whilst for *P. guajava* was 6.77E-03, 8.22E-03, & 6.55E-03 and for *A. digitata* were 2.35E-03, 1.81E-03, & 2.63E-03, in Kano, Jigawa and Borno state respectively. The estimated daily intake values for Cr were below the acceptable threshold

**Table 4. Descriptive statistics of the activity concentration of Cd in the medicinal plants from the three states.**

| | | Mean ± SD (mg/kg) | 95% Confidence Interval for Mean | | Minimum (mg/kg) | Maximum (mg/kg) |
|---|---|---|---|---|---|---|
| | | | Lower Bound (mg/kg) | Upper Bound (mg/kg) | | |
| C. papaya | Kano | 0.0051±0.00122 | 0.0045 | 0.0057 | 0.010 | 0.010 |
| | Jigawa | 0.0087±0.00008 | 0.0086 | 0.0087 | 0.010 | 0.010 |
| | Borno | 0.0064±0.00085 | 0.0060 | 0.0069 | 0.010 | 0.010 |
| | Total | 0.0067±0.0017 | 0.0063 | 0.0072 | 0.000 | 0.010 |
| P. guajava | Kano | 0.0149±0.00634 | 0.0114 | 0.0184 | 0.010 | 0.040 |
| | Jigawa | 0.0082±0.01326 | 0.0009 | 0.0155 | 0.000 | 0.060 |
| | Borno | 0.0176±0.0044 | 0.0152 | 0.0201 | 0.010 | 0.030 |
| | Total | 0.0136±0.00954 | 0.0107 | 0.0164 | 0.000 | 0.060 |
| A. digitata | Kano | 0.0092±0.00032 | 0.0090 | 0.0094 | 0.010 | 0.010 |
| | Jigawa | 0.0077±0.00011 | 0.0076 | 0.0077 | 0.010 | 0.010 |
| | Borno | 0.0084±0.00053 | 0.0081 | 0.0087 | 0.010 | 0.010 |
| | Total | 0.0084±0.00072 | 0.0082 | 0.0086 | 0.010 | 0.010 |

**Table 5. ANOVA analysis table for Cr.**

| Cr | | Sum of Squares | Df | Mean Square | F | Sig. |
|---|---|---|---|---|---|---|
| C. papaya | Between Groups | 0.000 | 2 | 0.000 | 190.683 | 0.000 |
| | Within Groups | 0.000 | 51 | 0.000 | - | - |
| | Total | 0.000 | 53 | - | - | - |
| P. guajava | Between Groups | 0.000 | 2 | 0.000 | 5.698 | 0.006 |
| | Within Groups | 0.001 | 42 | 0.000 | - | - |
| | Total | 0.001 | 44 | - | - | - |
| A. digitata | Between Groups | 0.000 | 2 | 0.000 | 243.154 | 0.000 |
| | Within Groups | 0.000 | 45 | 0.000 | | |
| | Total | 0.000 | 47 | - | - | - |

(MTDI) (mg/day) value (0.035–0.2) suggesting no significant health risk with respect to this metal.

In terms of Cd, EDI values observed were 1.49E-03, 2.52E-03, & 1.88E-03 for *C. papaya*, whilst for *P. guajava*, values were 4.34E-03, 2.39E-03, & 5.14E-04, and for *A. digitata* values of EDI were 2.68E-03, 2.24E-03, & 2.46E-03, in Kano, Jigawa and Borno state respectively. These values of EDI were also below the acceptable threshold (MTDI) (mg/day) value (0.02–0.07). Even though majority of elements presented no risk, accumulation of Cd in the body could pose considerable potential health concerns [44, 45]. The total estimated daily intake of all metals by an adult person from the ingestion of these herbs samples in the three states follow a sequence: Jigawa > Kano > Borno for (*C. papaya*), Kano > Jigawa > Borno for (*P. guajava*) and Jigawa > Borno > Kano for (*A. digitata*), whereas the corresponding amount from the ingestion of herbs (plants) was 0.177 mg/day/kg body weight. The estimated daily intake of Cr found from this investigation was averagely higher compared to Cd for the same ingestion rate

**Table 6. Multiple comparisons for Cr.**

| Dependent Variable | | (I) loc | (J) loc | Mean Difference (I-J) in (mg/kg) | Std. Error | Sig. | 95% Confidence Interval | |
|---|---|---|---|---|---|---|---|---|
| | | | | | | | Lower Bound (mg/kg) | Upper Bound (mg/kg) |
| C. papaya | LSD | kano | Jigawa | -0.00273333* | 0.000242 | 0.000 | -0.00322 | -0.00225 |
| | | | Borno | 0.00197778* | 0.000242 | 0.000 | 0.001491 | 0.002464 |
| | | Jigawa | kano | 0.00273333* | 0.000242 | 0.000 | 0.002247 | 0.00322 |
| | | | Borno | 0.00471111* | 0.000242 | 0.000 | 0.004225 | 0.005198 |
| | | Borno | kano | -0.00197778* | 0.000242 | 0.000 | -0.00246 | -0.00149 |
| | | | Jigawa | -0.00471111* | 0.000242 | 0.000 | -0.0052 | -0.00422 |
| P. guajava | LSD | kano | Jigawa | -0.00496667* | 0.001841 | 0.010 | -0.00868 | -0.00125 |
| | | | Borno | 0.000753 | 0.001841 | 0.685 | -0.00296 | 0.004469 |
| | | Jigawa | kano | 0.00496667* | 0.001841 | 0.010 | 0.001251 | 0.008683 |
| | | | Borno | 0.00572000* | 0.001841 | 0.003 | 0.002004 | 0.009436 |
| | | Borno | kano | -0.00075 | 0.001841 | 0.685 | -0.00447 | 0.002963 |
| | | | Jigawa | -0.00572000* | 0.001841 | 0.003 | -0.00944 | -0.00200 |
| A. digitata | LSD | kano | Jigawa | 0.00185000* | 0.000129 | 0.000 | 0.001591 | 0.002109 |
| | | | Borno | -0.00093750* | 0.000129 | 0.000 | -0.0012 | -0.00068 |
| | | Jigawa | kano | -0.00185000* | 0.000129 | 0.000 | -0.00211 | -0.00159 |
| | | | Borno | -0.00278750* | 0.000129 | 0.000 | -0.00305 | -0.00253 |
| | | Borno | kano | 0.00093750* | 0.000129 | 0.000 | 0.000678 | 0.001197 |
| | | | Jigawa | 0.00278750* | 0.000129 | 0.000 | 0.002528 | 0.003047 |

**Table 7. ANOVA analysis table for Cd.**

| Cd | | Sum of Squares | Df | Mean Square | F | Sig. |
|---|---|---|---|---|---|---|
| C. papaya | Between Groups | 0.000 | 2 | 0.000 | 77.393 | 0.000 |
| | Within Groups | 0.000 | 51 | 0.000 | - | - |
| | Total | 0.000 | 53 | - | - | - |
| P. guajava | Between Groups | 0.001 | 2 | 0.000 | 4.496 | 0.017 |
| | Within Groups | 0.003 | 42 | 0.000 | - | - |
| | Total | 0.004 | 44 | - | - | - |
| A. digitate | Between Groups | 0.000 | 2 | 0.000 | 69.042 | 0.000 |
| | Within Groups | 0.000 | 45 | 0.000 | - | - |
| | Total | 0.000 | 47 | - | - | - |

of the three medicinal plant types (Fig 1). However, a two-sample t-test was performed to know if there is a significant difference in the EDI values for Cd and Cr in mg/day/kg. We assumed the hypothesized mean difference (HMD) to be zero (mean values of the two groups (Cd & Cr) do not differ) and an α-value of 0.05. From the analysis, $p$ (one-tail) was obtained to be 0.12 which is greater than the α-value of 0.05. Hence the null hypothesis is accepted that there is no significant difference in the EDI (toxified with Cd & Cr). A further sample Cr and Cd independently t-test statistical analysis was performed with HMD of 0.045 mg/day/person for Cd and 0.1175 mg/day/person for Cr to know if the consumption rate was far above EDI upper limits. The $p$ (one-tail) = 3.10785E-14 for Cd and $p$ (one-tail) = 4.44826E-15 for Cr compare with the α-value of 0.05, this shows that EDI for both Cd & Cr is far below the upper acceptable limit.

**Table 8. Multiple comparisons table for Cd.**

| Dependent Variable | | (I) loc | (J) loc | Mean Difference (I-J) in (mg/kg) | Std. Error | Sig. | 95% Confidence Interval | |
|---|---|---|---|---|---|---|---|---|
| | | | | | | | Lower Bound (mg/kg) | Upper Bound (mg/kg) |
| C. papaya | LSD | Kano | Jigawa | -0.00353* | 0.00029 | 0.000 | -0.0041 | -0.0030 |
| | | | Borno | -0.00131* | 0.00029 | 0.000 | -0.0019 | -0.0007 |
| | | Jigawa | Kano | 0.00353* | 0.00029 | 0.000 | 0.0030 | 0.0041 |
| | | | Borno | 0.00222* | 0.00029 | 0.000 | 0.0016 | 0.0028 |
| | | Borno | Kano | 0.00131* | 0.00029 | 0.000 | 0.0007 | 0.0019 |
| | | | Jigawa | -0.00222* | 0.00029 | 0.000 | -0.0028 | -0.0016 |
| P. guajava | LSD | Kano | Jigawa | 0.00669* | 0.00323 | .0450 | 0.0002 | 0.0132 |
| | | | Borno | -0.00274 | 0.00323 | .4020 | -0.0093 | 0.0038 |
| | | Jigawa | Kano | -0.00669* | 0.00323 | .0450 | -0.0132 | -0.0002 |
| | | | Borno | -0.00943* | 0.00323 | .0060 | -0.0160 | -0.0029 |
| | | Borno | Kano | 0.00274 | 0.00323 | .4020 | -0.0038 | 0.0093 |
| | | | Jigawa | 0.00943* | 0.00323 | .0060 | 0.0029 | 0.0160 |
| A. digitata | LSD | Kano | Jigawa | 0.00152* | 0.00013 | 0.000 | 0.0013 | 0.0018 |
| | | | Borno | 0.00077* | 0.00013 | 0.000 | 0.0005 | 0.0010 |
| | | Jigawa | Kano | -0.00152* | 0.00013 | 0.000 | -0.0018 | -0.0013 |
| | | | Borno | -0.00074* | 0.00013 | 0.000 | -0.0010 | -0.0005 |
| | | Borno | Kano | -0.00077* | 0.00013 | 0.000 | -0.0010 | -0.0005 |
| | | | Jigawa | 0.00074* | 0.00013 | 0.000 | 0.0005 | .00010 |

**Table 9. The EDI of heavy metals in mg/day/kg body weight from the consumption of the medicinal herbs from Kano, Jigawa and Borno.**

| Medicinal plant | EDI Values (mg/day/kg) | | Total EDI through Consumption of the three medicinal herbs |
|---|---|---|---|
| | EDI (Cd) | EDI (Cr) | |
| C. papaya* | 1.49E-03 | 2.40E-03 | 3.90E-03 |
| P. guajava* | 4.34E-03 | 6.77E-03 | 1.11E-02 |
| A. digitata* | 2.68E-03 | 2.35E-03 | 5.03E-03 |
| C. papaya** | 2.52E-03 | 3.20E-03 | 5.72E-03 |
| P. guajava** | 2.39E-03 | 8.22E-03 | 1.06E-02 |
| A. digitata** | 2.24E-03 | 1.81E-03 | 4.05E-03 |
| C. papaya*** | 1.88E-03 | 1.82E-03 | 3.70E-03 |
| P. guajava*** | 5.14E-03 | 6.55E-03 | 1.17E-02 |
| A. digitata*** | 2.46E-03 | 2.63E-03 | 5.08E-03 |
| Maximum acceptable Daily Intake (MTDI) (mg/day) | 0.02–0.07[abc] | 0.035–0.2[ac] | |

[a] = [46],

[b] = [47],

[c] = [48]

* = Kanu,

** = Jigawa,

*** = Borno

## Potential health risks associated with long-term exposure

**The Target Hazard Quotient (THQ) and Health Index (HI).** The results of the THQ values for the MCCs in the medicinal plants are presented in Fig 2. Individual THQ values for extremely toxic metal Cd surpass 1 for all the medicinal plant leaf extracts under investigation except for P. quajava (leaves) from Borno. Also, the THQ for Cr was below 1 except for P quajava (leaves) in the three states (Borno, Jigawa and Kano). The combined contributions of the MCCs resulted in a THQ value greater than 1 for all the plants from the three states. The combined THQ values were observed to be high and may present detrimental health concerns throughout one's lifetime based on the MCCs contents alone. There is a clear indicator that consumption of medicinal herbs originating from Kano, Jigawa, and Borno states areas could instigate a health risk to the local population as well as those outside of the region owing to Cd as the THQ values > 1 due to *P. guajava* consumption. The collective effect of the absorption of highly toxic metals from the ingestion of various medicinal herbs from the three states was estimated through the calculation of hazard index (HI) as indicated in Eq. 3 and the results were 17.923 (Cd) and 12.619 (Cr). Generally, in agreement with the fact that these herbs accrue toxic metals to a larger degree than non-leafy plants, nearly 49.01% of the health index (HI) was associated with ingestion of *P. guajava* from Kano for Cd, whereas 60.94% of the HI (Fig 3) was also associated with ingestion of *P. guajava* from Kanu for Cr. In Borno state, 64% of HI was observed to be associated with *P. guajava* due to Cr, 54.12% of HI was associated *P. guajava* in Jigawa resulting from the ingestion of Cr and 48.38% of the HI (Fig 3) was observed in association with the ingestion of *A. digitata* in Borno state due to Cd. However, all the remaining values were below 35% of the HI associated with ingestion of *C. papaya* and *A. digitata* from all the three states.

**The Target Cancer Risk (TCR) and Cancer Risk (CR) assessment.** The MCCs are classified by the International Agency for Research on Cancer (IARC) as carcinogenic agents [48].

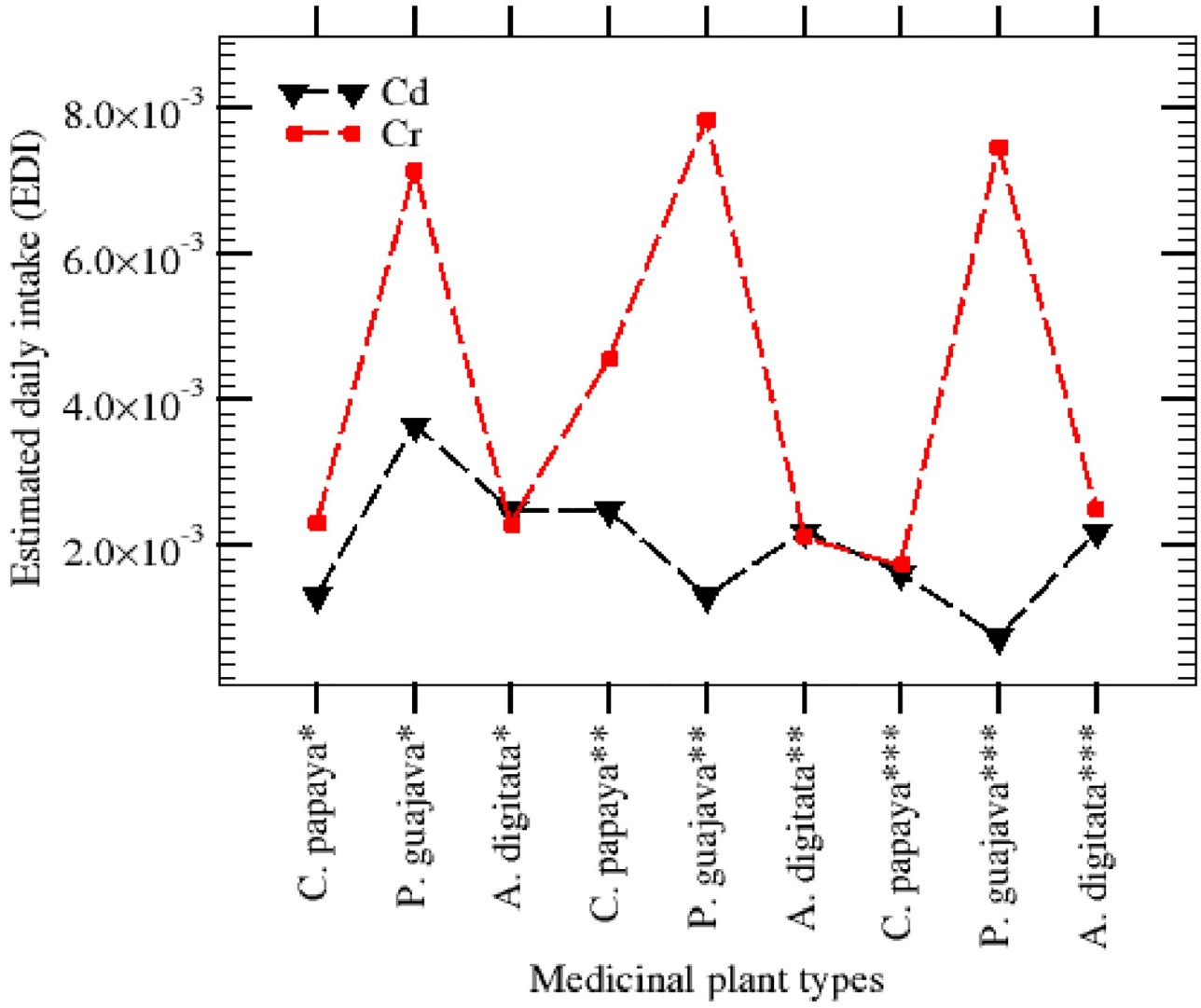

**Fig 1. The estimated daily intake of Cr and Cd in the three herbs from Kanu, Jigawa, and Borno states.**

Long-lasting exposure to low doses of the MCCs could therefore result in many types of cancers. The calculated cancer risk for the MCCs through the studied leaves of the medicinal plants from Borno, Kano and Jigawa States are presented in Fig 4. According to [32] the recommended safe limit of cancer risk resulting from MCCs in plant samples is set at $< 1$ in every 1,000,000 lifetime exposure (CR$< 10^{-6}$), and the threshold risk lime is set at CR$< 10^{-4}$, that is, a chance of $< 1$ in every 10,000 where remedial measures are considered. In the case where public health safety is considerable, a limit of $< 1$ in every 1,000 is considered. In this study, the CR for Cd violated the upper limit of $< 1$ in every 10, 000 ($< 10^{-4}$) in all the studied medicinal plants from the three states. This implies that the risk of cancer developing as a result of consuming these plants from the three states (Borno, Jigawa and Kano States) is high. Also, the cumulative risk of cancer (Fig 4) development due to the consumption of any of the studied medicinal plants leading to the accumulation of the MCCs exceeded the limit of $< 10^{-4}$. However, among all the studied medicinal plants: *P. guajava* had the highest chances of cancer risks (CR) for Cr of 1.25E-05, 1.18E-05, and 1.13E-05 from the three states: Jigawa,

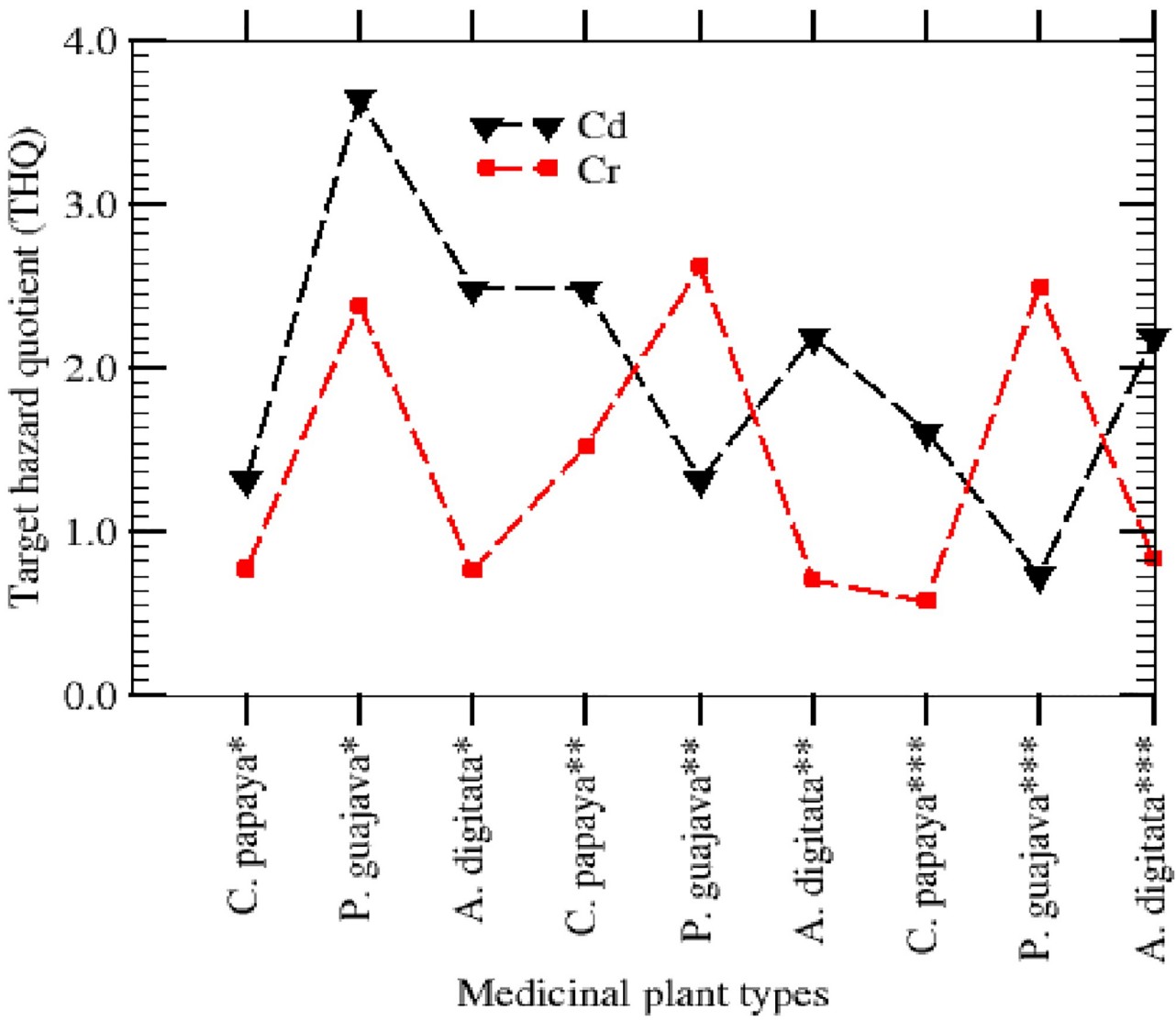

**Fig 2. Target hazard quotient due to consumption of three herbs from Kano, Jigawa, and Borno states.**

Borno and Kano, respectively. These cancer risk values due to the consumption of the *P. guajava* (guava) leaves show that 12 cancer cases may occur in every 10,000 people who are regularly consuming the plant in Jigawa State, and 11 in every 10,000 people in Borno and Kano States.

The TCR (Fig 3) values as a result of exposure to Cd and Cr through the ingestion of the herbs in all three states were 0.007 and 0.019 respectively. These values were beyond the maximum permissive threshold value of 0.0001, suggesting a significant risk of cancer in adult people from the ingestion of these herbs' types. This study has revealed that Cr is the most dominant carcinogen in all the three States and therefore, attention should be focused to control its exposure to the environment. A prompt resolution should be taken to control and possibly avoid the excessive use of these leaves for medicinal purposes within these three states since most of the people who used them dwell in rural areas.

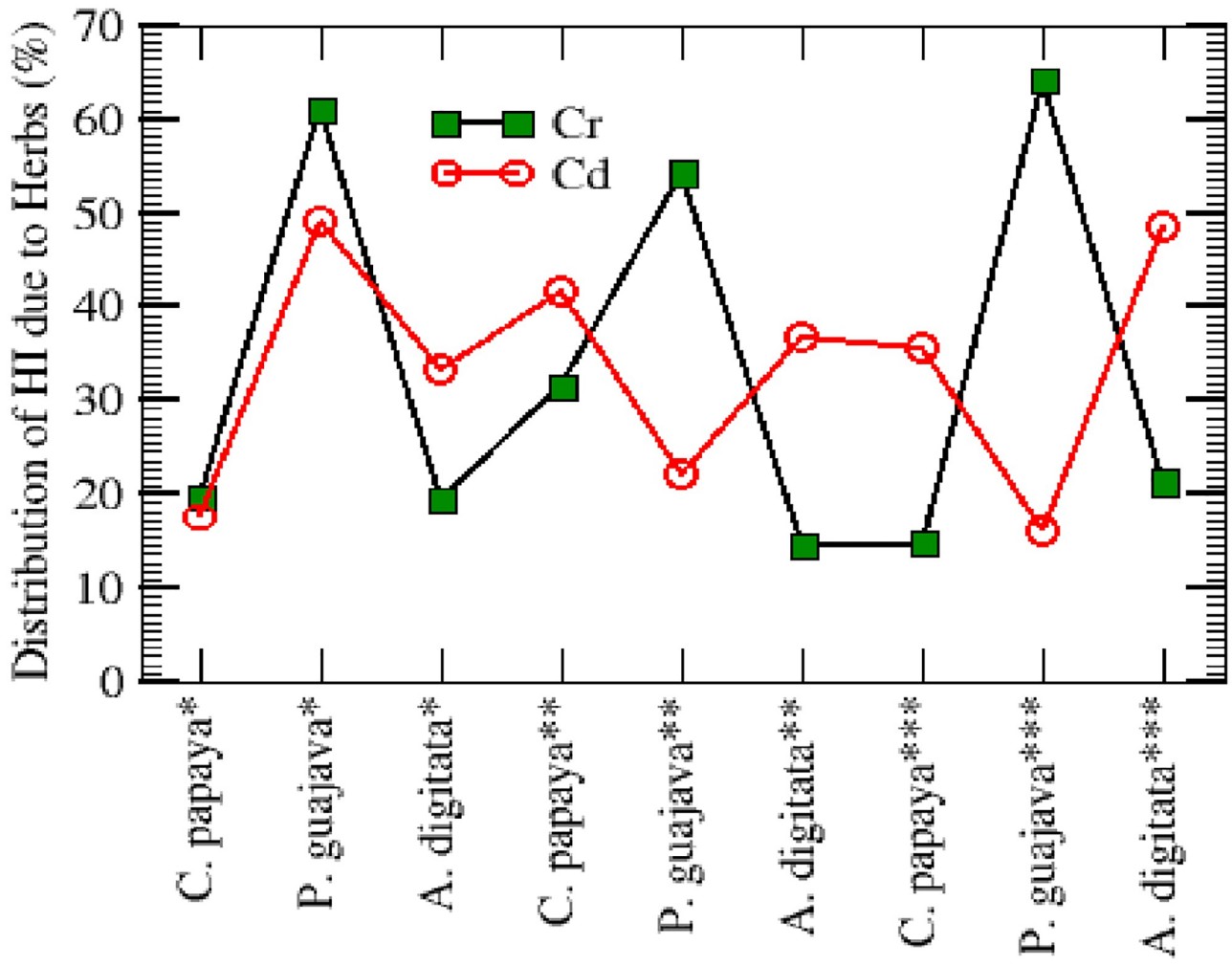

**Fig 3. Distribution in the percentage of HI associated with ingestion of herbs from Kano, Jigawa & Borno States.**

## Conclusion

This study evaluated the levels of hazardous metals in herbal extracts from Kano, Jigawa, and Borno states. The result obtained showed the presence of high levels of toxic metals in the three types of leaf extracts. The work revealed that was a statistically significant difference in Cr concentration between groups as shown by the one-way ANOVA with C. papaya ($F = 190.683$, $p = 0.000$), P. guajava ($F = 5.698$, $p = 0.006$), A. digitata ($F = 243.154$, $p = 0.000$). Post hoc test revealed that the C. papaya and A. digitata observed concentrations were statistically significant across the three states with ($p = 0.000$). There was no statistically significant difference between concentrations of the extracts between Kano and Borno states for P. guajava ($p = 0.686$). For Cd, the one-way ANOVA result for Cd is similar to that of Cr and there was a statistically significant difference in Cd concentration between groups as shown by the one-way ANOVA with C. papaya ($F = 77.393$, $p = 0.000$), P. guajava ($F = 4.496$, $p = 0.017$), A. digitata ($F = 69.042$, $p = 0.000$). Post hoc test with multiple comparison revealed that the activity concentration of all extracts was statistically significant across the three states with ($p < 0.05$).

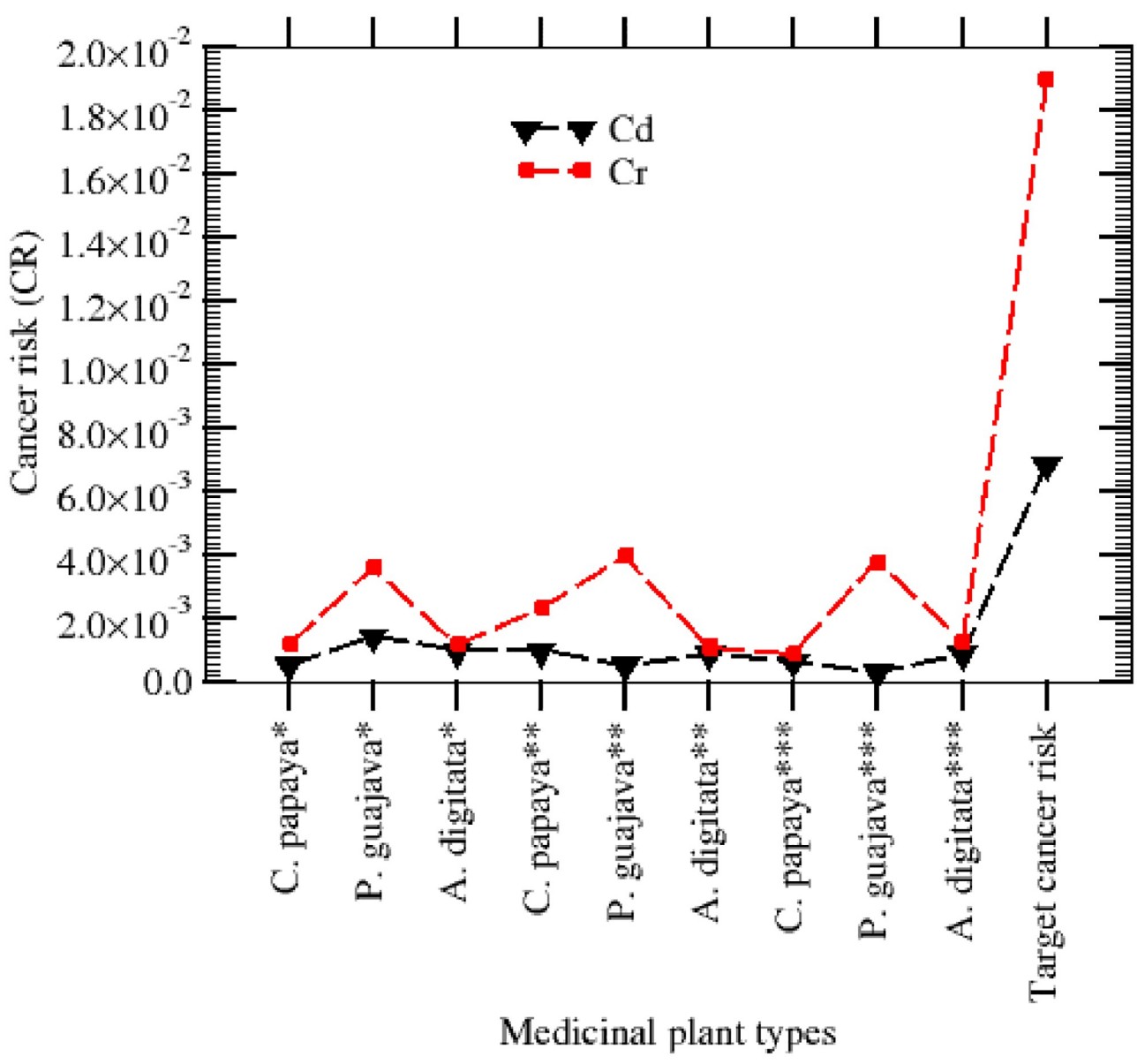

**Fig 4. The cancer risk and target cancer risk in ingesting these polluted herb types.**

All extracts from the three states were discovered to have EDIs values that were significantly lower than the maximum permitted daily consumption for each metal. However, the THQ of toxic metals from ingestion of *P. guajava* was observed to be > 1 for Cr and Cd due to the ingestion of all plant extracts. Also, the HI values exceeded one when separately ingesting all the plant extracts from the three states. The TCR revealed possible cancer risk due to the consumption of the plant extracts as evidenced by the corresponding TCR values of the indicated metals that exceed the maximum threshold value of 0.0001. Therefore, special attention should be paid to the safety of herbal extracts consumed and distributed by states to safeguard the welfare of the people in the region and beyond.

### Further research

Further research is important in this area to measure the MCCs deposition rate in various organs in our body and the mechanisms of the MCCs absorption/inclusion during the leaf juice extract production process. Also, the study of agrarian techniques should be able to reduce heavy metal translocation in the herbs. These works should involve the influence of herb varieties and seasonal variations, soil types, and local traditional customs.

## Supporting information

**S1 Raw images.**
(DOCX)

**S1 File.**
(DOCX)

## Author Contributions

**Conceptualization:** Raymond Limen Njinga.

**Formal analysis:** Raymond Limen Njinga, Adebiyi Samuel Adebayo.

**Methodology:** Raymond Limen Njinga.

**Software:** Raymond Limen Njinga.

**Writing – original draft:** Raymond Limen Njinga.

**Writing – review & editing:** Ayodele Philip Olufemi.

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
