## [Decision Letter · Decision Letter 0]

24 Jan 2022

PONE-D-21-39473MAJOR CHEMICAL CARCINOGENS AND HEALTH EXPOSURE RISK IN SOME THERAPEUTIC HERBAL PLANTS IN NIGERIAPLOS ONE

Dear Dr. Njinga,

Thank you for submitting your manuscript to PLOS ONE. After careful consideration, we feel that it has merit but does not fully meet PLOS ONE’s publication criteria as it currently stands. Therefore, we invite you to submit a revised version of the manuscript that addresses the points raised during the review process.

We look forward to receiving your revised manuscript.

Kind regards,

Noshin Ilyas

Academic Editor

PLOS ONE

Journal Requirements:

2. Thank you for including information about the identification of the samples used in this study. Please provide the name of the expert consulted in identifying the samples.

Additionally, in your Methods section, please provide additional information regarding the permits you obtained for the work. Please ensure you have included the full name of the authority that approved the field site access and, if no permits were required, a brief statement explaining why.

"No"

b) State what role the funders took in the study. If the funders had no role in your study, please state: “The funders had no role in study design, data collection and analysis, decision to publish, or preparation of the manuscript.

5. We note that Figure 1 in your submission contain copyrighted images. All PLOS content is published under the Creative Commons Attribution License (CC BY 4.0), which means that the manuscript, images, and Supporting Information files will be freely available online, and any third party is permitted to access, download, copy, distribute, and use these materials in any way, even commercially, with proper attribution. For more information, see our copyright guidelines: http://journals.plos.org/plosone/s/licenses-and-copyright.

Additional Editor Comments:

This paper needs comprehensive revisions, more details are required about collected samples, statistical analysis and work. Moreover, authors should add recent literature in introduction and discussion.

Reviewers' comments should also be properly addressed

Reviewers' comments:

Reviewer's Responses to Questions

**Comments to the Author**

1. Is the manuscript technically sound, and do the data support the conclusions?

Reviewer #1: Partly

Reviewer #2: Partly

2. Has the statistical analysis been performed appropriately and rigorously? 

Reviewer #1: No

Reviewer #2: No

3. Have the authors made all data underlying the findings in their manuscript fully available?

Reviewer #1: Yes

Reviewer #2: No

4. Is the manuscript presented in an intelligible fashion and written in standard English?

Reviewer #1: Yes

Reviewer #2: No

5. Review Comments to the Author

Reviewer #1: The manuscript entitled "MAJOR CHEMICAL CARCINOGENS AND HEALTH EXPOSURE RISK IN SOME

THERAPEUTIC HERBAL PLANTS IN NIGERIA" can be accepted with major revisions.

Methodology and result section is poorly written in non-technical way.

Results in table 2 reveal that, the amount of Cd in C. guajava collected from Kanu, and Jigwa was .037 and .05 mg/ kg which was little bit higher than WHO standards (.02 mg/kg); further its significance should be tested with z-test. On the other hand the Cr as depicted in table 2, is in safe-limits (less than 1.3 mg/kg); its significance can also be test by z-test.

In table 3, the EDI is also within the acceptable limits, so what is the author claiming for?

Further more, I would like to recommend some statistical analyses to make the manuscript findings better understandable.

There would be analysis of variance (ANOVA) using two factors (Locations and Plant-species) using HSD as post-hoc for each studied variable.

Strictly, there would be a z-test for EDI, THQ, HI, TCR in which author should compare the observed average values with standard acceptable values; the results would be presented in the form of z-values, and p-values for each variable. Without the z-test, the findings and claim of the author is NOT reliable.

Quality of graphs (data visualization) is poor can be made better by highlighting (horizontal line at y-axis parallel to x-axis) the acceptable ranges in each graph

Reviewer #2: Dear Authors,

The article, “MAJOR CHEMICAL CARCINOGENS AND HEALTH EXPOSURE RISK IN SOME THERAPEUTIC HERBAL PLANTS IN NIGERIA” highlights the issue contamination in herbal remedies. The heavy metals (Cd and Cr) are the major toxicants in plants (Herbs or Vegetables), and these metals are sourced from irrigation water or the polluted soil. However, as I reviewer I have some comments that need to be addressed before reaching any final decision.

First the abstract has to be revised thoroughly, the second sentence starting “Accumulation of MCCs….” Is confusing and need to be revised. I am wondering how the authors named heavy metals (Cd and Cr) as MCC?? Abstract has many typos, difficult to highlight each one here. It’s not clear which plant has higher concentration of these metals?

Why the authors have not considered investigating several other heavy metals? I wonder if the authors have also studied the nutritionally important metals in these plants?

The introduction part is weak, it needs improvement in terms of detailed information of C. papaya, P. guajava, and A. digitate. The introduction section just lists the review of literature it lacks the real on-ground information, socio-economic value of these plants? Also, It will be good to have pictures of these plants and the plant distribution in Nigeria, etc.

The presented data lack the proper statistical analysis. As per my understanding, the plant samples were collected from different locations in Nigeria, hence it will be best to point to these localities with a higher concentration of heavy metals in samples. This information will be good for the folks and herbalists to refrain to use these plants in their formulations.

The number of samples per plant sample is low I just wonder if the toxic metals are really sourced from plants or in solvents?? Did the Authors try to analyze these metals in the solvents?

6. PLOS authors have the option to publish the peer review history of their article (what does this mean?). If published, this will include your full peer review and any attached files.

Reviewer #1: No

Reviewer #2: No

---

## [Author Response · Author response to Decision Letter 0]

27 May 2022

The document to this effect has been attached.

---

## [Decision Letter · Decision Letter 1]

12 Aug 2022

PONE-D-21-39473R1MAJOR CHEMICAL CARCINOGENS AND HEALTH EXPOSURE RISK IN SOME THERAPEUTIC HERBAL PLANTS IN NIGERIAPLOS ONE

Dear Dr. Njinga,

Thank you for submitting your manuscript to PLOS ONE. After careful consideration, we feel that it has merit but does not fully meet PLOS ONE’s publication criteria as it currently stands. Therefore, we invite you to submit a revised version of the manuscript that addresses the points raised during the review process.

Please see the reviewers' comments below. Please note that if a reviewer has requested citations to specific articles, those articles should only be cited if they are directly relevant to the study. If you find that any number of the requested citations are irrelevant or inappropriate, please state this in your Response to Reviewers for each article you assess to be irrelevant or inappropriate to cite.

In addition, we ask you to respond fully to the following editorial matters. Please note that further consideration of your manuscript is contingent on satisfying these requirements:

1) Thank you for your response stating "We have cited the expert with similar medicinal plants in their articles." However, this does not address our request regarding how the samples used in this study were reliably identified. Please provide in your manuscript text the name of the expert consulted in identifying the samples, or otherwise please provide detailed information regarding how the samples were definitively identified.

2) Thank you for your response stating "No permit is needed since this is an individual research work and might be used by the government as baseline data." However, this does not address our request regarding permits obtained for the work. Specifically, we ask you to clarify in your manuscript text whether any permits (such as for site access) were obtained for the field work in gathering the samples used in this study. If no permits were required, please instead include in your manuscript text a brief statement explaining why.

3) During our internal evaluation of the manuscript, we found significant text overlap between your submission and the following previously published works:

"Vegetables contamination by heavy metals and associated health risk to the population in Koka area of central Ethiopia" (https://doi.org/10.1371/journal.pone.0254236)

Please revise the manuscript to rephrase the duplicated text, cite your sources, and provide details as to how the current manuscript advances on previous work. Please note that further consideration is dependent on the submission of a manuscript that addresses these concerns about the overlap in text with published work.

We look forward to receiving your revised manuscript.

Kind regards,

Hugh Cowley

Staff Editor

PLOS ONE

Journal Requirements:

Reviewers' comments:

Reviewer's Responses to Questions

**Comments to the Author**

1. If the authors have adequately addressed your comments raised in a previous round of review and you feel that this manuscript is now acceptable for publication, you may indicate that here to bypass the “Comments to the Author” section, enter your conflict of interest statement in the “Confidential to Editor” section, and submit your "Accept" recommendation.

Reviewer #1: All comments have been addressed

Reviewer #3: All comments have been addressed

2. Is the manuscript technically sound, and do the data support the conclusions?

Reviewer #1: Yes

Reviewer #3: Yes

3. Has the statistical analysis been performed appropriately and rigorously? 

Reviewer #1: Yes

Reviewer #3: Yes

4. Have the authors made all data underlying the findings in their manuscript fully available?

Reviewer #1: Yes

Reviewer #3: Yes

5. Is the manuscript presented in an intelligible fashion and written in standard English?

Reviewer #1: Yes

Reviewer #3: Yes

6. Review Comments to the Author

Reviewer #1: This is an interesting study and the authors have collected a unique dataset using cutting edge methodology. The paper is generally well written and structured

Reviewer #3: The revised paper can be accepted after Minor revision as suggested below

1) Statistical analysis details to be mentioned as footnote in each table

2) Some of the following recent and relevant publications can be cited. Its only suggestion.

Sustainability, 2021, 13, 8471. https://doi.org/10.3390/su13158471

International J of cancer management, 2021, 14(2):e89116 doi:10.5812/ijcm.89116

https://link.springer.com/chapter/10.1007/978-981-10-8548-2_19

Plos One, 2022, 17(4): e0266676. https://doi.org/10.1371/journal.pone.0266676

7. PLOS authors have the option to publish the peer review history of their article (what does this mean?). If published, this will include your full peer review and any attached files.

Reviewer #1: No

Reviewer #3: **Yes: **R.Z. Sayyed

---

## [Author Response · Author response to Decision Letter 1]

29 Sep 2022

We have reviewed the manuscript thoroughly.

---

## [Editor Report · Decision Letter 2]

6 Oct 2022

MAJOR CHEMICAL CARCINOGENS AND HEALTH EXPOSURE RISKS IN SOME THERAPEUTIC HERBAL PLANTS IN NIGERIA

PONE-D-21-39473R2

Dear Dr. Njinga,

We’re pleased to inform you that your manuscript has been judged scientifically suitable for publication and will be formally accepted for publication once it meets all outstanding technical requirements.

Kind regards,

Hugh Cowley

Staff Editor

PLOS ONE
---

## [Editor Report · Acceptance letter]

11 Oct 2022

PONE-D-21-39473R2 

Major chemical carcinogens and health exposure risks in some therapeutic herbal plants in nigeria 

Dear Dr. Njinga:

I'm pleased to inform you that your manuscript has been deemed suitable for publication in PLOS ONE. Congratulations! Your manuscript is now with our production department. 

Kind regards, 

on behalf of

Mr Hugh Cowley 

Staff Editor

PLOS ONE